# Rhabdomyolysis in Pediatric Patients with SARS-CoV-2 Infection

**DOI:** 10.3390/children9101441

**Published:** 2022-09-22

**Authors:** Ping-Sheng Wu, Shi-Bing Wong, Ching-Feng Cheng, Chun-Hsien Yu

**Affiliations:** 1Department of Pediatrics, Taipei Tzu Chi Hospital, Buddhist Tzu Chi Medical Foundation, New Taipei City 23142, Taiwan; 2School of Medicine, Tzu Chi University, Hualien City 97004, Taiwan

**Keywords:** COVID-19, rhabdomyolysis, child, hospital care

## Abstract

Background: Rhabdomyolysis is a rare but severe complication in adult patients with Coronavirus disease 2019 (COVID-19), which can result in acute kidney injury and death; however, it is rarely reported in pediatric patients. Methods: In this study, we retrospectively reviewed the clinical features and outcomes of rhabdomyolysis in pediatric patients aged 0–18 years with COVID-19 who were hospitalized at Taipei Tzu Chi Hospital, an epicenter of COVID-19 in northern Taiwan. Results: We treated eight patients with rhabdomyolysis during the omicron variant-Severe acute respiratory syndrome coronavirus 2 (omicron variant-SARS-CoV-2) community outbreak and none during the alpha variant endemic. These eight patients shared stereotypical presentations, including the presence of bilateral calf pain after defervescence. The creatinine kinase (CK) levels were between 1346 and 6937 U/L on admission, and clinical course was uneventful after aggressive saline hydration. Conclusion: Rhabdomyolysis is not a rare complication in pediatric patients with the omicron-SARS-CoV-2 infection, and reassurance of a good prognosis is important to alleviate family anxiety.

## 1. Introduction

Coronavirus disease 2019 (COVID-19) caused by Severe acute respiratory syndrome coronavirus 2 (SARS-CoV-2) has resulted in significant morbidity and mortality, primarily from severe pneumonia and the associated respiratory consequences. Severe muscle inflammation and injury have also been reported and can lead to multi-organ failure and death [1,2,3,4,5,6,7,8]. Rhabdomyolysis, defined by a serum creatine kinase (CK) greater than five times the upper limit of normal, or greater than 1000 U/L, was a common finding among hospitalized adult patients in 2020 when the SARS-CoV-2 alpha variant was spreading worldwide [9]. The need for renal replacement therapy as well as in-hospital mortality increase in adult patients with COVID-19 associated with rhabdomyolysis; however, rhabdomyolysis has rarely been reported for pediatric patients with COVID-19. The clinical course and prognosis of pediatric patients with COVID-19-associated rhabdomyolysis have not been determined, although some adolescent patients develop severe renal damage [10,11,12].

The mechanisms of how SARS-CoV-2 triggers muscle inflammation include direct viral entry into the muscle from a viral spike protein attached to the ACE-2 receptor and virus-triggered immune activation/inflammation [13]. The alpha-SARS-CoV-2 community outbreak occurred in northern Taiwan from May to July 2021, and it was controlled successfully by combined non-pharmaceutical interventions, including contact tracing, quarantine, universal facial masking, and school closures [14]. However, the omicron-SARS-CoV-2 outbreak started in northern Taiwan in March 2022 and spread quickly due to the “coexistence with the virus” policy of the Taiwanese government. Several pediatric patients were hospitalized at our institution due to rhabdomyolysis during the omicron-SARS-CoV-2 outbreak. The omicron variant of SARS-CoV-2 possesses notable changes in spike proteins compared to the previous alpha and delta variants, resulting in significantly different disease transmissibility, severity, and immune escape [15]. In this study, we describe the clinical features and outcomes of rhabdomyolysis in pediatric patients with COVID-19 by a retrospective chart review of admissions to Taipei Tzu Chi Hospital, an epicenter of COVID-19 in northern Taiwan.

## 2. Materials and Methods

This retrospective study was undertaken at Taipei Tzu Chi Hospital. Patients were identified by applying search criteria to the hospital’s electronic medical records registry. The inclusion criteria were patients of <18 years old with a positive result for SARS-CoV-2 by the reverse-transcription-polymerase chain reaction assay (RT-PCR) on a nasal swab sample, and were hospitalized to the ward or enhanced centralized quarantine center operated by Taipei Tzu Chi Hospital. We retrospectively collected information from two specific times of 28 May–28 July 2021, during the alpha-SARS-CoV-2 outbreak, and 1 April–10 June 2022, during the omicron-SARS-CoV-2 outbreak. Patients’ presenting symptoms, vital status, and inpatient laboratory data were manually collected from the medical records. This study was approved by the local Ethics Committee of Taipei Tzu Chi General Hospital (11-X-091). Written informed consent was waived because the study was a retrospective data analysis.

## 3. Results

A total of 104 patients of <18 years old were admitted to the ward or enhanced centralized quarantine center operated by Taipei Tzu Chi Hospital from 28 May to 28 July 2021. None of these patients reported calf pain during hospitalization, and no patient was diagnosed with rhabdomyolysis. In contrast, 243 patients of <18 years old with COVID-19 were hospitalized from 1 April to 10 June 2022, and 10 presented with myositis or rhabdomyolysis. Two patients were excluded from further analysis because of underlying Duchenne muscular dystrophy and a CK level of <1000 U/L. The remaining eight patients were six males and two females aged 5–10 years (Table 1). All eight patients had fever when COVID-19 was diagnosed and had an uneventful course before admission. They were all afebrile when they visited our clinic or emergency room. The chief complaint of these patients was bilateral calf pain which occurred 3–5 days after the fever started (Table 1), and the calf pain was so severe in five patients that they walked in antalgic gait or even refused to stand. One of them presented with kneeling down and moving on knees. The remaining three patients had calf pain and tenderness, but they could walk relatively normally. No obvious muscle weakness was observed in these eight patients, and they presented normal knee-jerk reflexes. Neither of them had muscular disease nor experienced myositis or rhabdomyolysis before this episode. Other neurological examinations in these patients were unremarkable. One patient received 10 μg BNT162b2 2 days before the SARS-CoV-2 infection, and the other seven patients did not receive vaccination. The SARS-CoV-2 RT-PCR cycle threshold (Ct) value of these patients was between 13 and 24 (Table 1). The CK levels were 1346–6937 U/L at admission. However, no CK-MB or troponin-I elevation was observed (Table 1). Renal function indicators, including blood urea nitrogen and creatinine, were within normal limits in all patients (Table 1). Five received a urinalysis, and all were negative for urine occult blood and red blood cells (Table 1). Relative leukopenia was frequently found in these patients with white blood cell counts of 2320–5350/μL, and white cell count was inversely associated peak CK concentration (Spearman’s rank correlation; ρ = −0.810; *p* = 0.015). All these patients had very low C-reactive protein (<0.1 mg/dL). The nasopharyngeal sample from one patient was tested by multiplex PCR and only SARS-CoV-2 was detected.

Fever was not detected in any of these patients during hospitalization. We treated them with aggressive intravenous normal saline hydration (1.2–1.5 times of daily maintenance fluid) for 2–3 days and acetaminophen or ibuprofen for pain relief as needed. None of them was prescribed with bicarbonate infusion. Calf pain relieved on day 2 after admission in all patients and no difficulty walking was reported. Serum CK levels during admission are illustrated in Figure 1. The CK concentration peaked on day 4 after the COVID-19 symptoms started in five, on day 6 in one, and day 7 in two patients (Figure 1). During hospitalization, no tea-colored urine was observed in these patients, and no renal replacement therapy was needed. All of these patients were discharged from the ward in 3–5 days without complications. 

## 4. Discussion

In our institution, we did not observe any pediatric patients with the alpha-SARS-CoV-2 infection who developed rhabdomyolysis. In contrast, eight children with COVID-19 were hospitalized because of rhabdomyolysis during the omicron variant-SARS-CoV-2 outbreak, which resulted in difficulty walking. These patients recovered quickly after treatment with intravenous fluids aggressively, and no renal complications were found. Our observations suggest that rhabdomyolysis is relatively frequent during the convalescent phase of the omicron variant-SARS-CoV-2 infection, and is generally benign and self-limited.

Two possible mechanisms of the action of SARS-CoV-2 on skeletal muscle have been proposed, including a direct mechanism by binding to the ACE2 receptor, and an indirect mechanism through an exaggerated inflammatory process in skeletal muscle [13]. However, the expression of the ACE-2 gene does not occur in human skeletal muscle according to a human single-cell RNA sequencing dataset, which makes direct muscle infection by SARS-CoV-2 less possible [13]. In line with the molecular modeling data, SARS-CoV-2 is not detected in muscle samples of patients who die of COVID-19 [16]. In our study, all patients developed rhabdomyolysis after defervescence, suggesting that the indirect mechanism is more likely the pathogenesis of SARS-CoV-2-related muscle involvement. In addition, our study illustrated an inverse correlation between WBC count and peak CK concentration, which further implied the involvement of post-infectious immune reaction in the pathogenesis of SARS-CoV-2-associated rhabdomyolysis. Many cytokines which were significantly increased during SARS-CoV-2 infection possibly induce muscle fiber proteolysis and promote a decrease in protein synthesis, such as interleukin-6 (IL-6), interleukin-1β (IL-1β), interleukin-8 (IL-8 or CXCL-8), interferon gamma (IFN-γ), interferon-gamma inducible protein 10 (IP-10 or CXCL10), and tumor necrosis factor alpha (TNF-α) [13]. Post-mortem psoas muscle histopathology also showed MHC-1 immunostaining of muscle fibers and CD68-positive, CD4-positive, and/or CD8-positive histiocytes and T cells infiltration [16]. Taken together, muscle damage induced by SARS-CoV-2 occurred secondary to an inflammatory response, including damage from cytokines.

Since the COVID-19 pandemic began, Taiwan has been extraordinarily successful in containing SARS-CoV-2 spread with a combination of case-based interventions (including testing, contact tracing, and quarantine) and population-based interventions (including physical distancing and facial masking with wide adherence) [17,18]. However, an alpha varinat-SARS-CoV-2 outbreak occurred between April and June 2021 in Taipei, which resulted in hundreds of COVID-19 related death [19,20]. The alpha varient-SARS-CoV-2 outbreak was successfully controlled through nationwide Level 3 epidemic prevention and control measures and school closures [21]. Since 7 August 2021, newly diagnosed domestic COVID-19 infection was found for less than 10 patients daily, indicating a near elimination of disease outbreak. Since the omicron variant-SARS-CoV-2 outbreak from April, 2022, the Taiwanese government changed the policy to “coexist with virus”; therefore, escalating COVID-19 cases reached more than 5 million from April to August in 2022 [22]. This specific alpha- and omicron variant-SARS-CoV-2 epidemiological pattern in Taiwan allowed a great chance to observe different clinical presentations of these SARS-CoV-2 variants. In our study, we observed eight patients with rhabdomyolysis associated with omicron-SARS-CoV-2 infection but none with alpha-SARS-CoV-2, and this observation provided clinical evidence of distinct immune responses toward different SARS-CoV-2 variants [23,24,25,26]. Compared to alpha-SARS-CoV-2, the omicron variant induced lower inflammatory responses and chemokine reactions in ACE-2-expressing mice [25] and humans [24]. The omicron variant also showed less suppression to the innate immune system in airway epithelial cells [26]. The attenuated immune responses of omicron-SARS-CoV-2 indeed correlated to its decreased disease severity [27,28]; however, these features contradicted the observation of the current study, indicating the increased occurrence of rhabdomyolysis by the omicron variant. Further immunologic and histopathologic studies are warranted to explore the pathomechanisms of rhabdomyolysis induced by SARS-CoV-2 viruses.

Infection has been proposed as the leading cause of pediatric rhabdomyolysis, and influenza-associated myositis may contribute to more than 80% of viral myositis cases [29]. Although only one patient in our study was checked for co-infecting pathogens, co-infection with influenza in these patients was unlikely because of the low prevalence of influenza in the community under the strict infection control strategies during the COVID-19 pandemic [30]. The clinical presentations of omicron variant-SARS-CoV-2-associated myositis in the current study, including a predominance of 5–9-year-old males, the presence of calf pain during the convalescent phase of febrile illness, and a self-limited benign course, were all comparable to typical influenza-associated myositis or benign acute childhood myositis (BACM) [29,31]. Similar to previous experience in BACM, no patients in our study presented myoglobinuria, which is a predictive factor of acute renal failure [31]. Therefore, for children with BACM, routinary examinations including serum CK, renal function and dipstick urinalysis, and outpatient care with analgesia and rest were sufficient [31]. In our opinion, omicron-SARS-CoV-2-associated rhabdomyolysis presented a typical clinical course as BACM. After routinary studies to exclude renal involvement, outpatient care and regular clinical follow-up are acceptable. However, our experience was contradicted by a recent systemic review of 86 patients, mostly adults, with SARS-CoV-2-associated rhabdomyolysis, reporting that 28% of patients required intravenous hemofiltration, 36% underwent mechanical ventilation, and 30% died [32]. There are several possible reasons for these results. First, omicron variant-SARS-CoV-2 causes mild clinical symptoms compared to the alpha variant, which was the predominant pathogen in that report [32]. Second, the outcome of rhabdomyolysis may be better in pediatric patients than in adults due to the lower risk of acute kidney injury. Third, case reports of SARS-CoV-2-induced rhabdomyolysis are particularly severe or unusual and led to an unfavorable prognosis [32]. Therefore, larger studies are warranted to clarify the incidence and prognosis of SARS-CoV-2-associated rhabdomyolysis in pediatric patients.

This study had several limitations. First, because of the strict quarantine policy during the alpha-SARS-CoV-2 outbreak in Taiwan, a large proportion of asymptomatic patients were hospitalized [14], which may have decreased the incidence of alpha-SARS-CoV-2-associated rhabdomyolysis. Second, our study was retrospectively designed with a relatively small sample size. Third, although we collected data from two distinct times during the alpha- and omicron-SARS-CoV-2 outbreaks, we did not have individual sequencing data to confirm the two variants. Fourth, only one patient in our study was evaluated for other pathogens that may have caused rhabdomyolysis; therefore, we cannot exclude the possibility of co-infection. Despite these limitations, this is the first study to illustrate a difference in the occurrence of rhabdomyolysis in patients with an alpha- and omicron-SARS-CoV-2 infection as well as the relatively benign course in pediatric patients. The different immune responses induced by these two variants in skeletal muscle are worthy of further study.

## 5. Conclusions

In conclusion, rhabdomyolysis should be considered in children who develop calf pain and difficulty walking after defervescence of an omicron-SARS-CoV-2 infection, and aggressive intravenous fluid treatment improved the condition quickly. The reassurance of a good prognosis of rhabdomyolysis in pediatric patients with COVID-19 is important to alleviate family anxiety.

## Figures and Tables

**Figure 1 children-09-01441-f001:**
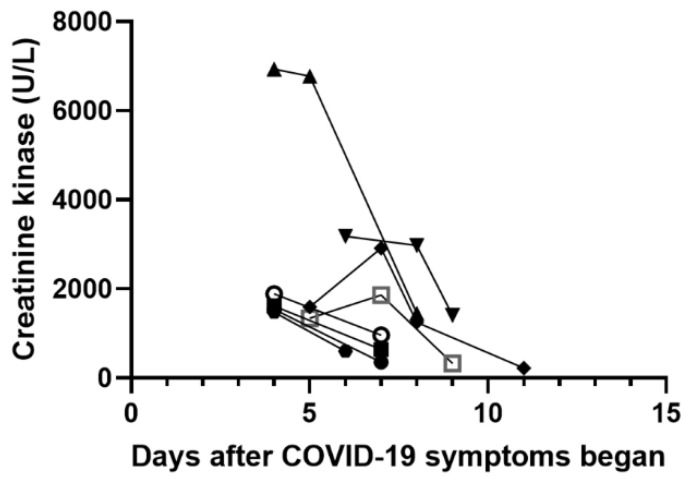
Changes in creatinine kinase (CK) in eight patients with rhabdomyolysis after COVID-19 symptoms began. The trends of individuals’ CK level were shown by the solid lines.

**Table 1 children-09-01441-t001:** Demographic and baseline laboratory data of pediatric patients with rhabdomyolysis after omicron-SARS-CoV-2 infection.

Patient Number	1	2	3	4	5	6	7	8
Age (years)	6	10	10	6	8	7	6	5
Sex	M	F	M	F	M	M	M	M
Duration between calf pain and fever (days)	5	5	4	3	5	4	4	4
SARS-CoV-2 RT-PCR CT value	16	17	20	24	13	15	21	17
White blood cell count (/μL)	3220	2320	4170	2650	3940	4500	5350	4710
Hemoglobin (g/dL)	13.1	13.0	14.7	12.6	13.2	14.0	11.6	13.6
Platelet count (×10^3^/μL)	227	201	250	231	121	231	178	277
Creatine kinase (CK, U/L)	1596	3180	1543	6937	1614	1892	1346	1475
Creatine kinase MB fraction (CK-MB, ng/mL)	19.8	66.8	47.5	91.0	20.9	30.1	18.2	19.2
Aspartate aminotransferase (AST, U/L)	ND	184	68	170	66	65	119	60
Alanine aminotransferase (ALT, U/L)	15	29	23	34	16	20	25	19
Blood urea nitrogen (BUN, mg/dL)	12	9	13	9	11	13	7	11
Serum creatinine (mg/dL)	0.46	0.45	0.46	0.28	0.38	0.49	0.38	0.31
C-reactive protein (CRP, mg/dL)	<0.1	<0.1	<0.1	<0.1	<0.1	<0.1	<0.1	<0.1
Serum sodium (mmol/L)	138	140	138	141	138	140	141	137
Serum potassium (mmol/L)	3.9	3.7	4.0	4.2	3.9	3.9	4.0	3.7
Urine occult blood	-	ND	-	-	ND	ND	-	-
Urine red blood cell count (/HPF)	0–2	ND	0–2	0–2	ND	ND	0–2	0–2

Abbreviation: F, female; HPF, high power field; M, male; ND, no data; RT-PCR, reverse-transcription-polymerase chain reaction.

## Data Availability

The data that support the findings of this study are available on request from the corresponding author, CHY. The data are not publicly available due to their containing information that could compromise the privacy of research participants.

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
