# Peer review of "Rhabdomyolysis in Pediatric Patients with SARS-CoV-2 Infection"

_children, 2022, doi:10.3390/children9101441_

Round 1
Reviewer 1 Report
I apologize for my late answer.
I read this paper with a great interest and don't have any particular comment.
In the reference number 4, I think there a small mistake : dos Santos = Dos Santos.
Reviewer 2 Report
Thanks to the authors for the opportunity of reviewing this interesting paper regarding occurrence of rhabdomyolisis after SARS-CoV-2 infection in pediatric patients.
The authors did not observe any pediatric patients with the alpha-SARS-CoV-2 infection who developed rhabdomyolysis. On the other side they found eight children hospitalized because of rhabdomyolysis during the omicron-SARS-CoV-2 outbreak. Clinical and laboratory features seem absolutely compatible with the well-known clinical entity "benign acute myositis" that is associated to viral infection (primarily influenza virus) and largely described in literature.
Considering that, i think it would be useful to expand the discussion on the comparison between rhabdomyolysis from SARS-CoV-2 and other viruses.
There are clinical management and differential diagnosis protocols for benign acute myositis in literature. Do you believe that in SARS-CoV-2-related myositis these can still be valid or do you propose different approach?
Could you better specify the neurological findings of clinical examination? How were tendon reflexes? What do you mean with difficulty walking? complete refusal of the load? walking on toes?other?
Did you administer any analgesics for pain control?
Considering that benign acute myositis may be recurrent in the same patient, are there anyone in your case series who has had a previous episode of rhabdomyolysis?
Reviewer 3 Report
The authors present an interesting analysis regarding muscular effects of SARS-CoV-2 infection, which has been rarely discussed in literature. The paper has a weak point (i.e. the low number of cases), properly underlined by the authors themselves, but on the other hand it's original.
Just some minor reviews are needed:
1) specify abbreviations within the abstract;
2) specify SARS-CoV-2 abbreviation in line 25;
3) substitute "COVID-19" with "SARS-CoV-2" in line 115 (patients die due to COVID-19 or they die due to SARS-CoV-2 infection).
Round 2
Reviewer 2 Report
Thanks to the authors for clearly addressing all my questions.
I am satisfied with all the changes made.